# Prevalence and aeroallergen sensitization in pediatric Allergic Rhinitis: A population-based study in Jeju, Korea

Min-su Oh[1,2�], Miok Kim[3�], Jung-Kook Song[2,4*], Seong-Chul Hong[2,4]

1 Department of Pediatrics, Jeju National University College of Medicine, Jeju, Republic of Korea, 2 The Environmental Health Center, Jeju, Republic of Korea, 3 Department of Internal Medicine, Jeju National University College of Medicine, Jeju, Republic of Korea, 4 Department of Preventive Medicine, Jeju National University College of Medicine, Jeju, Republic of Korea

he These authors contributed equally to this work.
* salbab@hotmail.com

## Abstract

Allergic rhinitis (AR) is a prevalent chronic disease among children, influenced by environmental and genetic factors. Understanding regional aeroallergen sensitization patterns is essential for targeted intervention. This study investigates the prevalence of AR and its associated aeroallergen sensitization among schoolchildren in Jeju, Korea. A cross-sectional, population-based study was conducted in 2016 on 1,067 schoolchildren aged 9–16 years. Data were collected using the International Study of Asthma and Allergies in Childhood (ISAAC) questionnaire and skin prick tests for 18 common aeroallergens. Statistical analysis was performed using multiple logistic regression to determine risk factors for AR. The prevalence of AR was 29.0%, with the highest sensitization rates observed for Dermatophagoides pteronyssinus (76.5%), Dermatophagoides farinae (60.9%), and Japanese cedar pollen (39.1%). Sensitization to multiple aeroallergens (polysensitivity) was significantly associated with increased AR risk (OR 1.58, 95% CI: 1.017–2.455, p = 0.042). Age-related variations in sensitization patterns were noted, with younger children showing higher overall sensitization rates. This study provides evidence of region-specific allergen sensitization patterns in pediatric AR. The high prevalence of Japanese cedar pollen sensitization highlights the need for further investigation into environmental exposure and potential preventive measures.

## Introduction

Allergic rhinitis (AR) is a major public health concern, affecting 10% to 40% of populations worldwide [1]. Its prevalence has risen globally in recent decades [2,3], particularly in regions with previously low or moderate prevalence, while stabilizing or even declining in regions with the highest prevalence [4–6]. AR typically manifests

**Data availability statement:** All relevant data are within the manuscript and its Supporting Information files.

**Funding:** The author(s) received no specific funding for this work.

**Competing interests:** The authors have declared that no competing interests exist.

in childhood, with about 80% of cases occurring before age 20: 20% by age 2–3, around 40% by age 6, and 30% during adolescence [7]. AR is classified into seasonal, perennial, and perennial with seasonal exacerbations. The onset and resolution of symptoms are closely linked to the environmental concentration of seasonal or perennial allergens. Even after exposure to indoor and outdoor allergens ceases, persistent symptoms can be triggered by smoking, other pollutants, noxious odors, temperature change, exercise and psychological stress [1]. Diagnosis is based on clinical history and physical examination with sensitization confirmed through skin prick tests or serologic analyses of IgE antibodies [8].

Allergen sensitization is a key risk factor for AR, yet its patterns are shaped by complex interactions between genetic and environmental factors. Consequently, there is no strict dose-response relationship between allergen exposure and allergic sensitization, and the specific allergens to which an individual will become sensitized remain unpredictable [9]. While national surveys have provided valuable insights into sensitization patterns [10–15], national-level sensitization data may have limited applicability in smaller, environmentally distinct populations compared to large-scale, national populations. Jeju Island exemplifies this issue. Despite comprising only 1.27% of Korea's total population (approximately 65 million in Jeju out of 51 million nationwide as of 2017 [16]), Jeju reports the highest AR prevalence among the nation's 17 administrative districts [17]. This discrepancy may be attributed to island's distinctive environmental conditions, shaped by its dominant agricultural industry. The island is known for its extensive tangerine orchards, which are commonly surrounded by Japanese cedar (*Cryptomeria japonica*, JC) trees as shelterbelts. This long-standing practice has led to significantly higher airborne concentrations of JC pollen compared to other regions [18], suggesting that the sensitization profile of Jeju's population may differ from national trends. Given these unique environmental factors, it is crucial to investigate whether Jeju's pediatric population exhibits distinct sensitization patterns compared to national data.

This study aims to assess the prevalence of AR and characterize the aeroallergen sensitization patterns among schoolchildren in Jeju. By addressing gaps in regional data, this research underscores the limitations of applying national sensitization trends to smaller, environmentally distinct populations and highlights the need for localized epidemiologic assessments to better inform public health strategies.

## Materials and methods

### Study population

Jeju Island is geographically divided by Mt. Halla into two main administrative regions: Jeju City in the north and Seogwipo City in the south. Historically, human settlements have developed along coastal areas where spring water sources (Yongcheonsu) are available. Currently, approximately 75% of Jeju's population resides in Jeju City in the north. Considering the geographic balance, we aimed to include schools evenly distributed across the northern and southern regions. Subjects were ultimately recruited from schools that were cooperative and agreed to participate in our research.

A total of 1,918 subjects were recruited from 13 public schools in Jeju, South Korea, between 17/05/2016 and 09/06/2016, and they completed the ISAAC questionnaire. Among them, 1,131 students participated in skin testing. 64 students were excluded due to invalid test results. The study protocol was approved by the institutional review board of the Jeju National University Hospital (JEJUNUH2016-04-014), in accordance with the Declaration of Helsinki. Each student's parents provided written informed consent. Data were collected in a manner that maintained the confidentiality of the children's records.

## Assessment

We used the Korean version of the written questionnaire of ISAAC [19], which was distributed by teachers at the participating schools. Prior to completing the questionnaire, written informed consent was obtained from the parents of all participants. The AR based on the ISAAC questionnaire was defined as a positive answer to the following question: "In the past 12 months, has your child had a problem with sneezing or a runny or blocked nose when he/she did not have a cold or the flu?" After completion, the questionnaires were returned to the schools for collection. Questionnaires were filled out **by** parents for elementary school students and directly by students themselves for middle and high school participants [19]. Prick-puncture allergy skin testing was applied to the volar aspect of the forearm. The 18 common aeroallergens (*D. pteronyssinus, D.farinae*, JC, pine, willow, maple, birch, oak, alder, timothy grass, Bermuda grass, ragweed, mugwort, Japanese hop, *Penicillium, Aspergillus, Cladosporium,* and *Alternaria*) and 2 controls (positive and negative) were administered to all subjects. Each of the allergens was manufactured by Allergopharma (Reinbek, Germany) except JC. JC pollen (Greer Laboratories Inc., Lenoir, NC, USA) was diluted with 0.9% saline to a protein concentration of 100µg/mL, and the same volume of 50% glycerin was added. Lengths and widths of wheals were measured 15 minutes following application. Sensitization to an antigen was defined as a mean wheal size that was the same size as or larger than that of the positive control (allergen/histamine ratio ≥1). A skin test was considered invalid if the wheal size for the positive control (1% histamine) was smaller than 2 mm or if a wheal was observed in the negative control (normal saline 0.9%) [20].

## Statistical analysis

To clarify the prevalence of sensitization to each of the test aeroallergens, the percentages of the schoolchildren with positive skin test responses were estimated among the total subjects as well as among the schoolchildren with positive skin test responses to one or more aeroallergens. The number of positive skin test responses was also estimated as the percentage of the subjects among the sensitized subjects. The relationship between the prevalence of the sensitization pattern to JC pollen and school levels (age) was examined using the Cochran-Armitage test. The influence of age (school level), sex, dichotomous variable of sensitization to dust mites, JC pollen, or molds, and degree of sensitization (mono-sensitized or polysensitized) on the prevalence of AR was evaluated using univariate and multivariate logistic regression analyses. The variables included in the multivariate logistic regression analysis were selected based on clinical relevance, previous literature indicating their association with allergic rhinitis (AR), and the results from the univariate analysis (variables with a p-value < 0.25). Specifically, we included age group, sex, sensitization to dust mites, JC pollen, moulds, and the degree of sensitization.

Statistical analyses were conducted with IBM SPSS version 24.0 (IBM Co., Armonk, NY, USA) software. A significant difference was defined as P < 0.05.

## Results

### Participant characteristics

A total of 1,067 students were included in the final analysis (mean age, 12.5±2.6 years; median age, 13.0; interquartile range [IQR], 10–16). The number of girls was similar to the number of boys (male: female ratio, 1.00:1.08) among the

1,067 students. Among those, 473 (44%) subjects were aged 9–11 years and came from 6 elementary schools, 322 (30%) subjects were aged 13–14 years and came from 3 middle schools, and 272 (26%) subjects were aged 16 years and came from 4 high schools.

**Prevalence of positive skin test responses to aeroallergens**

Approximately half of the schoolchildren (51.5%, 550 of 1,067) had positive test responses to one or more aeroallergens. The highest prevalence was observed for dust mite (*D. pteronyssinus* and *D. farinae)* and JC pollen, while the lowest prevalence was recorded for pine (Table 1).

The highest number of positive skin test responses observed was 12. However, the most common number of positive skin test responses was two (34.9% of sensitized schoolchildren), followed by a single positive response (26.2). Only 4.8% of sensitized schoolchildren showed positive resposes to more than six aeroallergens. The mean and median numbers of positive test responses among sensitized schoolchildren were 2.5 and 2.0, respectively (Fig 1).

**Table 1. Prevalence of positive skin test responses to 18 aeroallergens.**

| Allergen tested | n | Among the total schoolchildren (n = 1,067) | Among the schoolchildren with positive test responses (n = 550) |
|---|---|---|---|
| One or more allergens | 550 | (51.5) | (100.0) |
| *D.pteronyssinus* | 421 | (39.5) | (76.5) |
| *D.farinae* | 335 | (31.4) | (60.9) |
| Japanese cedar | 215 | (20.1) | (39.1) |
| Pine | 4 | (0.4) | (0.7) |
| Willow | 21 | (2.0) | (3.8) |
| Maple | 23 | (2.2) | (4.2) |
| Birch | 22 | (2.1) | (4.0) |
| Oak | 22 | (2.1) | (4.0) |
| Alder | 28 | (2.6) | (5.1) |
| Timothy grass | 87 | (8.2) | (15.8) |
| Bermuda grass | 34 | (3.2) | (6.2) |
| Ragweed | 14 | (1.3) | (2.5) |
| Mugwort | 31 | (2.9) | (5.6) |
| Japanese hop | 35 | (3.3) | (6.4) |
| *Penicillium* | 6 | (0.6) | (1.1) |
| *Aspergillus* | 10 | (0.9) | (1.8) |
| *Cladosporium* | 16 | (1.5) | (2.9) |
| *Alternaria* | 75 | (7.0) | (13.6) |

Table in parentheses are percentages.

### Prevalence of AR

A total of 1,067 schoolchildren who completed both the questionnaire and allergy skin test were classified into four categories based on their sensitization status (+ or -) and the presence of AR symptoms (+ or -). Among them, 29.0% (309) were sensitized to one or more aeroallergens and had symptoms suggestive of AR in the past 12 months. These children were classified as having AR. Additionally, 22.6% (241) were sensitized but asymptomatic., 16.2% (173) had AR symptoms without allergic sensitization, and 32.2% (344) were non-allergic (Fig 2).

The prevalence of AR was 29.0% (309 of 1,067) among all schoolchildren, 56.2% (309 of 550) among sensitized children, and 64.1% (309 of 482) among children classified as AR cases based on the ISAAC questionnaire (Fig 3).

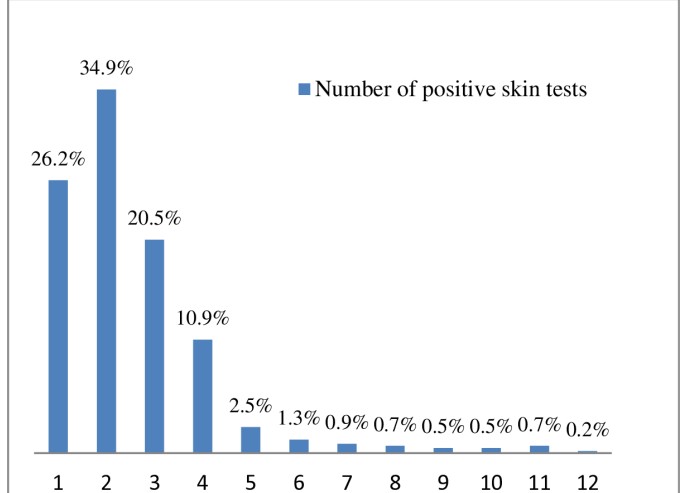

**Fig 1. Proportion of sensitized schoolchildren by the number of positive skin test responses.**

|  | ISAAC (+) | ISSAC (-) |  |
|---|---|---|---|
| **Skin prick test (+)** | 309 (29.0%) | 241 (22.6%) | 550 (51.6%) |
| **Skin prick test (-)** | 173 (16.2%) | 344 (32.2%) | 517 (48.4%) |
| **Total** | 482 (45.2%) | 585 (54.8%) | 1,067 (100%) |

**Fig 2. Categorization of schoolchildren using allergy skin test responses and the ISAAC questionnaire (n = 1,067).**

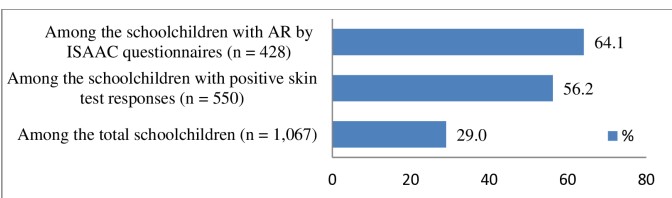

**Fig 3. Comparison of AR prevalence across three groups: total schoolchildren, sensitized schoolchildren, and AR cases based on the ISAAC questionnaire.**

## Sensitization patterns among AR

Among the 309 schoolchildren classified as having AR, three distinct sensitization patterns to JC pollen were identified. Sensitization to JC pollen only was observed in 5.5%, sensitization to JC pollen along with other aeroallergens in 34.0%, and sensitization to aeroallergens excluding JC pollen in 60.5%.

Significant associations were found between JC pollen sensitization patterns and age group (Table 2).

The prevalence of sensitization to JC pollen along with other aeroallergens increased significantly with school level (p < 0.001, Cochran-Armitage test): 22.6% (32 of 142) in elementary school, 41.1% (37 of 90) in middle school, and 46.8% (36 of 77) in high school. In contrast, the prevalence of sensitization to aeroallergens excluding JC pollen significantly decreased with school level (p < 0.001, Cochran-Armitage test): 73.9% (105 of 142) in elementary school, 53.3% (48 of 90) in middle school, and 44.2% (34 of 77) in high school. Although the prevalence of sensitization to JC pollen only appeared to increase with school level (3.5% in elementary, 5.6% in middle and 9.0% in high school), the trend was not statistically significant (p = 0.088, the Cochran-Armitage test).

## Risk factors of AR

Table 3 presents the characteristics associated with AR. In the univariate analysis, sensitization to dust mites was a significant risk factor for AR (OR: 1.66; (95% CI: 1.064–2.580); p = 0.025). The degree of sensitization was also a significant risk factor, with polysensitization showing a higher likelihood of AR (OR: 1.83; (95% CI: 1.247–2.687); p = 0.002). In the multivariate analysis, the degree of sensitization remained significantly and independently associated with AR risk (adjusted OR: 1.58; (95% CI: 1.017–2.455); p = 0.042).

Adjusted for age group, sex, sensitization to dust mites, sensitization to JC pollen, sensitization to moulds, degree of sensitization in the table.

## Discussion

This study provides important epidemiologic insights into the prevalence and sensitization patterns of allergic rhinitis (AR) among schoolchildren in Jeju, a region characterized by distinct environmental allergen exposure. Using the ISAAC questionnaire and skin prick tests for 18 common aeroallergens, we identified a sensitization rate of 51.5% among schoolchildren aged 9–16 years. The most prevalent allergens were *D. pteronyssinus*, *D. farinae*, and JC pollen, confirming that dust mites and JC pollen are dominant sensitizing agents in this population [21].

Our findings suggest that the overall sensitization rate of 51.5% among Jeju schoolchildren may be an underestimate due to the limited number of allergens included in the skin prick test panel. Notably, *D. farinae* was included, but two of the three most common allergens identified in the 2010 nationwide survey on allergic diseases in Korea (*D. farinae*, dog, and German cockroach) were absent [15]. Given the prevalence of these allergens in previous studies, the actual sensitization rate may be higher than reported. Furthermore, data from the Third National Health and Nutrition Examination Survey (NHANES III) in the United States (1988–1994) indicated a sensitization rate of 55.5% among children aged 10–19 years

**Table 2. Sensitization patterns to Japanese cedar pollen among allergic rhinitis by three school levels.**

|  | Total |  | Elementary |  | Middle |  | High |  | p-value |
|---|---|---|---|---|---|---|---|---|---|
|  | 309 | (100.0) | 142 | (100.0) | 90 | (100.0) | 77 | (100.0) |  |
| JCP only | 17 | (5.5) | 5 | (3.5) | 5 | (5.6) | 7 | (9.0) | 0.088 |
| JCP and others | 105 | (34.0) | 32 | (22.6) | 37 | (41.1) | 36 | (46.8) | < 0.001 |
| JCP negative | 187 | (60.5) | 105 | (73.9) | 48 | (53.3) | 34 | (44.2) | < 0.001 |

Numbers in parentheses are percentages. P-values were obtained by the Cochran-Armitage test.

Table 3. Crude and adjusted ORs from the multivariate logistic regression model for factors associated with AR among the sensitized subjects (n = 550).

| | Crude OR (95% CI) for AR | | p-value | Adjusted OR (95% CI) for AR | | p-value |
|---|---|---|---|---|---|---|
| Age group (school level) | | | 0.323 | | | 0.347 |
| 9-11 (elementary) | 1.00 | (reference) | | 1.00 | (reference) | |
| 13-14 (middle) | 0.85 | (0.570-1.278) | | 0.82 | (0.541-1.231) | |
| 16 (high) | 0.73 | (0.484-1.103) | | 0.74 | (0.479-1.134) | |
| Sex | | | 0.329 | | | 0.453 |
| Female | 1.00 | (reference) | | 1.0 | (reference) | |
| Male | 1.18 | (0.844-1.661) | | 1.14 | (0.808-1.610) | |
| Sensitization to dust mites | | | **0.025** | | | 0.219 |
| No | 1.00 | (reference) | | 1.0 | (reference) | |
| Yes | 1.66 | (1.064-2.580) | | 1.38 | (0.826-2.307) | |
| Sensitization to JC pollen | | | 0.831 | | | 0.736 |
| No | 1.00 | (reference) | | 1.0 | (reference) | |
| Yes | 1.04 | (0.735-1.467) | | 1.07 | (0.729-1.564) | |
| Sensitization to moulds | | | 0.277 | | | 0.389 |
| No | 1.00 | (reference) | | 1.0 | (reference) | |
| Yes | 1.29 | (0.816-2.033) | | 1.23 | (0.767-1.978) | |
| Degree of sensitization | | | **0.002** | | | **0.042** |
| Monosensitized | 1.00 | (reference) | | 1.0 | (reference) | |
| Polysensitized | 1.83 | (1.247-2.687) | | 1.58 | (1.017-2.455) | |

Allergic Rhinitis (AR), Odds Ratio (OR) and 95% Confidence Intervals (CI).

[12]. Racial and ethnic differences in sensitization patterns may also contribute to variations in prevalence estimates. NHANES III data demonstrated that individuals categorized as "other" ethnicity, which likely includes Korean ethnicity, had significantly higher odds of positive skin test responses (OR 1.5, 95% CI: 1.2–2.0, p < 0.001) compared to non-Hispanic white individuals [12]. These findings highlight the complexity of interpreting sensitization prevalence across different populations and underscore the potential for underestimation in our study.

A key finding was that 29.0% of the total sample (309 of 1,067) met the criteria for AR, a prevalence higher than those reported in previous Korean studies, such as Kwon et al. (23.3%) [22], Hahm et al. (22.1%) [23], and Kim et al. (20.8%) [14]. Among sensitized children, 56.2% (309 of 550) had AR, a rate slightly lower than the 63.5% reported in a Japanese adult population [24]. The highest prevalence of AR (64.1%) was observed among ISAAC-defined AR cases, reinforcing previous findings that questionnaire-based assessments may overestimate AR prevalence due to the inclusion of non-allergic rhinitis cases [25–27].

Comparative analyses further underscore the distinct sensitization profile in Jeju. In a Japanese study (2006–2007), JC pollen was reported as the predominant allergen (56%) [24]. Another Japanese study (2017–2019) similarly found JC pollen to be the most prevalent allergen, with a sensitization rate of 78.5% among 13-year-old adolescents with current rhinitis living in Tokyo [28]. In contrast, the sensitization pattern in Jeju was dominated by dust mites (*D. pteronyssinus* and *D. farinae*), while JC pollen ranked third.

In our study, JC pollen sensitization did not emerge as an independent risk factor for allergic rhinitis (AR). Although the sensitization rate to house dust mites was approximately twice as high as that to JC pollen, neither dust mite nor JC pollen sensitization remained significant independent predictors after multivariate adjustment. Instead, polysensitization to multiple aeroallergens was the only significant factor associated with AR risk.

These findings suggest that in regions like Jeju, the cumulative allergenic burden—rather than sensitization to a single specific allergen—plays a more critical role in AR development. The differences observed between our results and Japanese studies may reflect variations in environmental exposure, timing of sensitization, host genetic factors, and allergen profiles specific to each region. Moreover, regions with similar latitude and climatic conditions, such as Kyushu in Japan, may also exhibit allergen sensitization patterns influenced by subtropical environmental factors. A previous study reported that cedar and cypress pollen counts were associated with the prevalence of allergic diseases among Japanese schoolchildren [29], suggesting that pollen exposure patterns in southern Japan could provide a useful comparison for understanding regional differences in allergen sensitization. Further research is warranted to clarify the mechanisms underlying these regional differences and to better understand the role of polysensitization in the natural course of allergic diseases.

The observed age-related decrease in allergic rhinitis (AR), particularly after excluding Japanese cedar (JC) pollen sensitization, can be explained by multiple interrelated factors. First, differences in lifestyle and environmental exposure play a role: younger children generally spend more time indoors and in bed, resulting in greater exposure to indoor allergens such as house dust mites (HDM). As children grow older and engage in more outdoor activities, their exposure to indoor allergens decreases while exposure to outdoor allergens like JC pollen increases. Second, age-related changes in immune responses may contribute; immune systems in young children are still maturing, and early hypersensitivity to HDM may evolve into tolerance or shift as the immune system develops. Third, the natural progression of allergic diseases, often referred to as the "allergic march," suggests that symptoms of HDM-induced allergic rhinitis may naturally improve or even remit over time. Children sensitized to HDM at an early age often experience a reduction in symptoms as they grow, which could partially explain the observed decrease in HDM-related AR prevalence among older age groups.

Furthermore, a previous Korean study involving 2,991 schoolchildren with AR also identified *D. pteronyssinus* (76.8%) and *D. farinae* (68.1%) as the leading allergens, followed by birch pollen (10.8%) [14]. These regional differences highlight the necessity of considering geographic and climatic factors when interpreting allergen sensitization data.

In particular, Jeju's increasingly subtropical climate may partly explain the predominance of perennial indoor allergens such as house dust mites, which resemble sensitization patterns observed in tropical or subtropical regions. For instance, a large cross-sectional study in Mexico reported that 87% of allergic rhinitis patients in tropical zones were sensitized to dust mites, with far lower pollen sensitization compared to subtropical and temperate areas [30]. Similarly, in Asia, dust mite sensitization affects up to 90% of atopic individuals, with *Blomia tropicalis* and *Tyrophagus putrescentiae* frequently observed, particularly in humid urban areas [31]. A study in Cartagena, Colombia found that specific IgE to *D. pteronyssinus* (64.6%), *D. farinae* (74.7%), *E. maynei* (68.7%), and *B. tropicalis* (54.5%) was significantly associated with acute asthma in a tropical setting [32]. These findings suggest that Jeju's sensitization profile may share commonalities with tropical regions, underscoring the need for context-specific approaches to allergen monitoring and intervention.

Our study also identified significant age-related trends in JC pollen sensitization. The proportion of children sensitized to both JC pollen and other aeroallergens increased with school level (p<0.001) [22,33,34]. This pattern aligns with prior research suggesting that pollen sensitization accumulates over time due to prolonged exposure [11,14]. Although the prevalence of JC pollen-only sensitization appeared to increase with age (3.5% in elementary, 5.6% in middle, and 9.0% in high school), this trend did not reach statistical significance (p=0.088) [24]. However, Yoshida et al. [29] suggested that children in regions exposed to JC pollen may develop sensitization progressively until reaching a plateau. Similarly, Ozasa et al. [34] demonstrated that JC pollen sensitization increased with age among schoolchildren aged 6 to 14 years, and Sakashita et al. [24] observed a significant age-dependent rise in JC pollen sensitization among Japanese adults aged 20 to 49 years.

Interestingly, the prevalence of sensitization to aeroallergens excluding JC pollen was inversely associated with school level. Although aeroallergen sensitization typically increases in the second and third decades of life [11], this decreasing trend may reflect a sharply rising rate of dust mite sensitization at younger ages. Early childhood sensitization is often dominated by indoor allergens such as dust mites. Ozasa et al. [34] reported that recent birth cohorts exhibited stronger dust mite sensitization (HDM IgE > 17.5 UA/ml), attributing this trend to changes in domestic environments and lifestyle.

NHANES data further support this shift, showing that allergen sensitivities among Americans increased significantly between NHANES II (1976–1980) and NHANES III (1988–1994), with the largest increases observed in pediatric populations aged 6–9 years [10]. Given the abrupt "Westernization" of Jeju, including increased time spent indoors in well-insulated buildings with reduced ventilation [23,35], early-life exposure to dust mites may be playing a more dominant role in allergic sensitization. Additionally, environmental changes such as decreased microbial diversity and increased exposure to combustion by-products and endocrine-disrupting chemicals may further contribute to these trends [9].

Among asymptomatic children in our study (n = 585), 41.2% (241 of 585) were already sensitized to one or more aeroallergens. The Global Asthma and Allergy European Network (GA(2)LEN) study in 14 European countries demonstrated that individuals with allergen sensitization, whether symptomatic or not, had a significantly higher risk of developing AR or asthma [36]. This underscores the importance of monitoring sensitized individuals, as asymptomatic sensitization may progress to clinical disease over time.

Regarding a preventive intervention for AR, it seems reasonable to target the factors for which there is evidence of their effect. Allergic sensitization is fundamental in the development of allergic disease, and it is worthwhile to know which allergen sensitization affects AR prevalence. In the current study, sensitization to dust mites was a significant risk factor for AR prevalence in the univariate analysis but was not observed in the multivariate analysis (Table 3). It has been suggested that AR is mostly associated with outdoor allergens. In the case of perennial symptoms, however, AR can also be relevant to indoor allergens [37]. Salo et al. [38] reported that sensitization to dust mites did not remain significantly associated with current allergies or current hay fever, with adjustment for age, sex, ethnicity, education, poverty, body mass index, serum cotinine level and seven IgE clusters, in NHANES 2005–2006. They suggested that this finding contrasted with NHANES II (1976–1980), reflecting differences in age distribution, analytic methods, or both. It might also be likely that mite-sensitive AR had greater compliance with dust mite avoidance measures. On the other hand, Kanazawa et al. [39] showed that sensitization to dust mites was very strongly associated with JC pollen sensitization and concluded that elevated serum IgE was a consequence of specific sensitization to dust mites.

From a public health perspective, our findings emphasize the need for region-specific allergen monitoring and tailored prevention strategies. While polysensitization was a significant risk factor for AR (aOR: 1.58; (95% CI: 1.017–2.455); p = 0.042) [10], the role of specific allergens, particularly JC pollen, warrants further investigation. Existing recommendations for AR prevention, such as minimizing early-life dust mite exposure [9,40,41] and avoiding JC pollen during peak seasons [18,21], remain crucial. However, given the increasing prevalence of AR in Jeju and the potential impact of environmental changes, further studies are needed to explore the effects of urbanization, indoor air quality, and climate shifts on allergen sensitization [9,35]. Additionally, primary prevention strategies such as exclusive breastfeeding for at least six months, delayed introduction of solid foods, and avoidance of tobacco smoke exposure should continue to be reinforced [42,43].

This study has several limitations. As a cross-sectional study, it cannot establish causality between allergen exposure and AR development. Additionally, reliance on questionnaire-based definitions introduces the possibility of recall bias [44], and the skin prick test is subject to observer variability [45]. However, the use of a relatively comprehensive allergen panel enhances the reliability of our sensitization estimates. To build upon these findings and address current limitations, future research could benefit from incorporating additional data sources, such as pet allergen sensitization (cats and dogs), detailed environmental exposure measurements (e.g., indoor pollutants and ambient air pollution), and household socioeconomic factors. Including these variables may offer a more comprehensive understanding of the complex interplay between environmental conditions and AR prevalence among children in Jeju.

## Conclusions

This study highlights the unique allergen sensitization profile of Jeju schoolchildren and underscores the need for region-specific public health interventions. Given that small populations with distinct sensitization patterns are often

overlooked in national surveys, this research bridges the gap between national-level data and regional variations, emphasizing the need for localized allergen monitoring and tailored health strategies. Future longitudinal research is essential to understand the long-term trends in allergen exposure and AR prevalence in this population.

## Supporting information

**S1 Table. Questionnaire for rhinitis symptoms from the Korean International Study of Asthma and Allergies in Childhood questionnaire.**
(DOCX)

## Acknowledgments

The authors are very grateful to all children, parents, and school teachers for their cooperation and participation. We also sincerely thank the former and current presidents of the Environmental Health Center (Atopic Dermatitis and Allergic Rhinitis) at Jeju National University School of Medicine—Seong-Chul Hong, M.D., Ph.D.; Keun-Hwa Lee, Ph.D.; and Ju Wan Kang, M.D., Ph.D.—for their leadership and support in project administration. Additionally, we appreciate Jeong Hong Kim, M.D., Ph.D., and Hye-Sook Lee, Ph.D., for their contributions to data collection.

## Author contributions

**Conceptualization:** Min-su Oh, Miok Kim, Jung-Kook Song, Seong-Chul Hong.

**Data curation:** Jung-Kook Song.

**Formal analysis:** Jung-Kook Song.

**Methodology:** Jung-Kook Song.

**Visualization:** Jung-Kook Song.

**Writing – original draft:** Min-su Oh, Jung-Kook Song.

**Writing – review & editing:** Miok Kim, Jung-Kook Song.

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
