## [Decision Letter · Decision Letter 0]

18 Mar 2025

Dear Dr. Song,

Thank you for submitting your manuscript to PLOS ONE. After careful consideration, we feel that it has merit but does not fully meet PLOS ONE’s publication criteria as it currently stands. Therefore, we invite you to submit a revised version of the manuscript that addresses the points raised during the review process.

We look forward to receiving your revised manuscript.

Kind regards,

Sethu Thakachy Subha, M.S

Academic Editor

PLOS ONE

Journal Requirements:

3. Please remove all personal information, ensure that the data shared are in accordance with participant consent, and re-upload a fully anonymized data set.

Reviewers' comments:

Reviewer's Responses to Questions

**Comments to the Author**

1. Is the manuscript technically sound, and do the data support the conclusions?

Reviewer #1: Yes

Reviewer #2: Yes

Reviewer #3: Yes

2. Has the statistical analysis been performed appropriately and rigorously?

Reviewer #1: Yes

Reviewer #2: Yes

Reviewer #3: Yes

3. Have the authors made all data underlying the findings in their manuscript fully available?

Reviewer #1: Yes

Reviewer #2: Yes

Reviewer #3: No

4. Is the manuscript presented in an intelligible fashion and written in standard English?

Reviewer #1: Yes

Reviewer #2: Yes

Reviewer #3: Yes

Reviewer #1: The data unique to Korea and Jeju were interesting and original. I think it can be a better report if we supplement the data we collect a little more. The conclusions were drawn according to the data, and there is no statistical problem.

Reviewer #2: Dr. Song and colleagues reported on the prevalence and aeroallergen sensitization in pediatric allergic rhinitis based on a population-based study in Jeju, Korea.

In epidemiological research, unique regional allergy data are valuable as they provide important insights into the onset and increase of allergies. This study uses data from eight years ago, but it would be helpful to explain why it was published at this timing.

The results and discussion include findings on Japanese cedar; however, the sensitization rate appears to be lower than that reported in Japanese epidemiological studies. Please consider citing the following paper and discussing the differences:

Kiguchi T, et al. Pollen-food allergy syndrome and component sensitization in adolescents: A Japanese population-based study. PLoS One. 2021 Apr 14;16(4):e0249649.

Line 34: Since only the odds ratio is presented, please add the 95% confidence interval.

Line 50-51: Please remove the terms "in vitro" and "in vivo."

Line 57: The term "smaller" is unclear—please specify what it is being compared to.

Line 78: Please add the study design at the beginning of the Methods section.

Line 131: Should this be a "Table" instead of a "Figure"?

Reviewer #3: NO COMMENTS

**Do you want your identity to be public for this peer review?** For information about this choice, including consent withdrawal, please see our Privacy Policy

Reviewer #1: No

Reviewer #2: No

Reviewer #3: No

---

## [Author Response · Author response to Decision Letter 1]

27 Mar 2025

Relevant files have been uploaded.

---

## [Decision Letter · Decision Letter 1]

15 Apr 2025

Dear Dr. Song,

Thank you for submitting your manuscript to PLOS ONE. After careful consideration, we feel that it has merit but does not fully meet PLOS ONE’s publication criteria as it currently stands. Therefore, we invite you to submit a revised version of the manuscript that addresses the points raised during the review process.

We look forward to receiving your revised manuscript.

Kind regards,

Sethu Thakachy Subha, M.S

Academic Editor

PLOS ONE

Journal Requirements:

Reviewers' comments:

Reviewer's Responses to Questions

**Comments to the Author**

Reviewer #1: All comments have been addressed

Reviewer #3: All comments have been addressed

2. Is the manuscript technically sound, and do the data support the conclusions?

Reviewer #1: Yes

Reviewer #3: Yes

3. Has the statistical analysis been performed appropriately and rigorously?

Reviewer #1: Yes

Reviewer #3: Yes

4. Have the authors made all data underlying the findings in their manuscript fully available?

Reviewer #1: Yes

Reviewer #3: Yes

5. Is the manuscript presented in an intelligible fashion and written in standard English?

Reviewer #1: Yes

Reviewer #3: Yes

Reviewer #1: I think it's data with unique characteristics of Jeju area. Thank you.

Let me ask you a few questions.

1. In JC exclusion, AR decreased with age. Is this explained only by the effect of early sensitization of HDM?

2. Why doesn't JC sensitization come out statistically in the AR risk evaluation? JC is likely to be the most important AR risk in Japan, so why is the difference?

3. I'm hoping to add animals (dogs, cats), cockroaches, and Cypress to the SPT.

4. In the geographical data comparison, what if reference goes into places similar to Jeju in the latitude (especially places like Kyushu in Japan).

Thank you for your efforts.

Reviewer #3: no comments .

**Do you want your identity to be public for this peer review?** For information about this choice, including consent withdrawal, please see our Privacy Policy

Reviewer #1: No

Reviewer #3: No

---

## [Author Response · Author response to Decision Letter 2]

21 Apr 2025

Rebuttal Letter (the Second Review)

Dear Reviewer 1,

We sincerely appreciate you for the thoughtful and constructive second-round review. Below are our detailed responses to each question and suggestion:

1. "In JC exclusion, AR decreased with age. Is this explained only by the effect of early sensitization of HDM?"

Thank you for this insightful question. Indeed, the observed age-related decrease in allergic rhinitis (AR), specifically in Japanese cedar (JC) pollen exclusion, can be explained by multiple interrelated factors:

• Differences in lifestyle and environmental exposure: Younger children generally spend more time indoors and in bed, leading to increased exposure to house dust mites (HDM). As children age and engage in more outdoor activities, exposure to indoor allergens decreases while exposure to outdoor allergens such as JC pollen increases.

• Age-related changes in immune response: Immune systems in younger children are still maturing, and early hypersensitivity to HDM may evolve as tolerance develops or immune responses change with age.

• Natural progression of allergic rhinitis: The "allergic march" concept explains that allergic rhinitis symptoms may naturally improve or even remit over time. Children sensitized to house dust mites (HDM) in early childhood often experience symptom reduction as they age, which could partially explain the observed decrease in HDM-induced AR prevalence among older age groups.

We have now clarified and elaborated on these factors explicitly in the discussion section of the manuscript. (line 258~270)

The observed age-related decrease in allergic rhinitis (AR), particularly after excluding Japanese cedar (JC) pollen sensitization, can be explained by multiple interrelated factors. First, differences in lifestyle and environmental exposure play a role: younger children generally spend more time indoors and in bed, resulting in greater exposure to indoor allergens such as house dust mites (HDM). As children grow older and engage in more outdoor activities, their exposure to indoor allergens decreases while exposure to outdoor allergens like JC pollen increases. Second, age-related changes in immune responses may contribute; immune systems in young children are still maturing, and early hypersensitivity to HDM may evolve into tolerance or shift as the immune system develops. Third, the natural progression of allergic diseases, often referred to as the "allergic march," suggests that symptoms of HDM-induced allergic rhinitis may naturally improve or even remit over time. Children sensitized to HDM at an early age often experience a reduction in symptoms as they grow, which could partially explain the observed decrease in HDM-related AR prevalence among older age groups.

2. "Why doesn't JC sensitization come out statistically in the AR risk evaluation? JC is likely to be the most important AR risk in Japan, so why is the difference?"

Thank you for this important question. Although JC pollen is recognized as a major contributor to AR in Japan, our results suggest that in Jeju, the cumulative allergenic burden (i.e., polysensitization) plays a more critical role in AR development than sensitization to a single allergen. While the sensitization rate to house dust mites was approximately twice as high as that to JC pollen in Jeju (table 1), neither dust mites nor JC pollen remained significant independent predictors after multivariate adjustment. Instead, polysensitization to multiple aeroallergens emerged as the only significant risk factor (table 3).

Furthermore, to assess whether multicollinearity between house dust mite (HDM) sensitization and JC pollen sensitization might have affected the statistical results, we examined Cramér's V coefficient between these variables, which was 0.19, indicating a weak association. In addition, the variance inflation factors (VIFs) for the six independent variables included in the multivariate model ranged from 1.003 to 1.388, suggesting that multicollinearity was not a concern. Therefore, the lack of statistical significance for JC pollen sensitization in the multivariate analysis was not attributable to multicollinearity.

These findings suggest that, in regions like Jeju, the cumulative allergenic burden — rather than sensitization to a single specific allergen — plays a more critical role in AR development. The differences observed between our results and Japanese studies may reflect variations in environmental exposure, timing of sensitization, host genetic factors, and allergen profiles specific to each region. Further research is needed to clarify the mechanisms underlying these regional differences and to better understand the role of polysensitization in the natural course of allergic diseases.

We have added this point to the revised Discussion section. (line 241~250 and 255~270)

In our study, JC pollen sensitization did not emerge as an independent risk factor for allergic rhinitis (AR). Although the sensitization rate to house dust mites was approximately twice as high as that to JC pollen, neither dust mite nor JC pollen sensitization remained significant independent predictors after multivariate adjustment. Instead, polysensitization to multiple aeroallergens was the only significant factor associated with AR risk.

These findings suggest that in regions like Jeju, the cumulative allergenic burden—rather than sensitization to a single specific allergen—plays a more critical role in AR development. The differences observed between our results and Japanese studies may reflect variations in environmental exposure, timing of sensitization, host genetic factors, and allergen profiles specific to each region.

Further research is warranted to clarify the mechanisms underlying these regional differences and to better understand the role of polysensitization in the natural course of allergic diseases.

3. "I'm hoping to add animals (dogs, cats), cockroaches, and Cypress to the SPT."

We appreciate the reviewer’s suggestion to include common animals (dogs, cats), cockroaches, and cypress in the SPT panel. In fact, a similar point was raised by another reviewer during the first round of peer review, and we have already incorporated this consideration into our revised manuscript. Specifically, we added a discussion in the Discussion section (line 345~351) noting that our skin prick test panel was limited and did not include certain prevalent allergens.

This study has several limitations. As a cross-sectional study, it cannot establish causality between allergen exposure and AR development. Additionally, reliance on questionnaire-based definitions introduces the possibility of recall bias [43], and the skin prick test is subject to observer variability [44]. However, the use of a relatively comprehensive allergen panel enhances the reliability of our sensitization estimates. To build upon these findings and address current limitations, future research could benefit from incorporating additional data sources, such as pet allergen sensitization (cats and dogs), detailed environmental exposure measurements (e.g., indoor pollutants and ambient air pollution), and household socioeconomic factors. Including these variables may offer a more comprehensive understanding of the complex interplay between environmental conditions and AR prevalence among children in Jeju.

4. "In the geographical data comparison, what if reference goes into places similar to Jeju in the latitude (especially places like Kyushu in Japan)?"

Thank you for this excellent suggestion. We have now included comparative references to geographical regions with similar latitudes and environmental conditions, such as Kyushu, Japan, in the discussion section.

29. Yoshida K, Adachi Y, Akashi M, Itazawa T, Murakami Y, Odajima H, et al. Cedar and cypress pollen counts are associated with the prevalence of allergic diseases in Japanese schoolchildren. Allergy. 2013;68(6):757–763. doi:10.1111/all.12158

Moreover, regions with similar latitude and climatic conditions, such as Kyushu in Japan, may also exhibit allergen sensitization patterns influenced by subtropical environmental factors. A previous study reported that cedar and cypress pollen counts were associated with the prevalence of allergic diseases among Japanese schoolchildren [29], suggesting that pollen exposure patterns in southern Japan could provide a useful comparison for understanding regional differences in allergen sensitization.

Your valuable insights and detailed comments have significantly improved the quality of our manuscript. We greatly appreciate your time, effort, and expertise in helping us strengthen our work.

We hope that these revisions address all of the reviewer’s concerns.

We would also like to sincerely request your consideration regarding the urgency of our situation. Some of the co-authors of this manuscript are currently facing institutional transitions, and if the acceptance of this paper is not confirmed by May 8, their affiliation contracts will unfortunately be terminated. Given these circumstances, we humbly ask for your kind assistance in expediting the review process, if at all possible.

We are deeply grateful for the editor's and reviewers' valuable time and thoughtful feedback, which have greatly improved the quality of our manuscript. We remain fully committed to making any additional revisions immediately if needed.

Thank you very much for your kind understanding and generous support.

Sincerely,

Dr. Song and colleagues

---

## [Editor Report · Decision Letter 2]

23 May 2025

Prevalence and aeroallergen sensitization in pediatric allergic rhinitis: A population-based study in Jeju, Korea

PONE-D-25-06116R2

Dear Dr. Song,

We’re pleased to inform you that your manuscript has been judged scientifically suitable for publication and will be formally accepted for publication once it meets all outstanding technical requirements.

Kind regards,

Sethu Thakachy Subha, M.S

Academic Editor

PLOS ONE
---

## [Editor Report · Acceptance letter]

PONE-D-25-06116R2

PLOS ONE

Dear Dr. Song,

I'm pleased to inform you that your manuscript has been deemed suitable for publication in PLOS ONE. Congratulations! Your manuscript is now being handed over to our production team.

Kind regards,

on behalf of

Dr. Sethu Thakachy Subha

Academic Editor

PLOS ONE